# The application of virtual teams in the improvement of enterprise management capability from the perspective of knowledge transfer

**Anqi Zhang**👤*

Department of Business Administration, Shanghai University of International Business and Economics, Shanghai, China

* stephy89dd@126.com

## Abstract

To clarify the problems existing in the process of knowledge transfer of virtual teams, the team cooperation optimization model is combined with the technological innovation capability under the neural network (NN) model, and it is expected to provide a reference for the knowledge management and technological innovation of enterprises. This study first explores the organization, collaboration, and management of virtual teams from the perspective of knowledge transfer. In addition, based on NN, the research and development (R&D) and innovation capabilities of enterprises are studied, and an evaluation index system is constructed. In the stage of empirical analysis, this study selects Enterprise A for research and analyzes the status quo and existing problems of this company. Then, this study proposes a set of schemes conducive to virtual team collaboration and comprehensively uses management theory to evaluate the application effect. The team collaboration model based on knowledge transfer is applied to the operation and management of the enterprise, which significantly improves the collaboration between the various departments of the enterprise and the fluidity of resources. An enterprise R&D capability evaluation system based on a NN model can comprehensively evaluate various index data, thereby prompting its further strategic management and investment. The virtual team collaboration model combined with the R&D technological innovation under the NN model can objectively evaluate the capabilities of the enterprise, thereby improving the management efficiency and benefiting the long-term development of the enterprise.

## 1. Introduction

With the development of Internet information technology, the production and lifestyle of human society have undergone tremendous changes. While information technology brings convenience, traditional concepts and models are also subtly replaced by innovative thinking models. Among them, the most obvious is the popularity and application of information

**Data Availability Statement:** All relevant data are within the manuscript and its Supporting Information files.

**Funding:** This study was supported by Shanghai Philosophy and Social Science Planning Youth

Project, "Research on Health Information Avoidance Behavior in Epidemic Prevention and Control," in the form of funds to AZ [2020EGL003].

**Competing interests:** The authors have declared that no competing interests exist.

virtualization [1]. In the context of the era of the knowledge economy, the investment and share of knowledge and information resources have gradually increased. The comprehensive and efficient use of information resources can bring maximum economic benefits to enterprises [2]. A virtual team is a new form of work organization based on information technology. It brings together individuals with common goals or common interests to form a team. In a narrow sense, virtual teams exist only in the virtual network world. In a broad sense, virtual teams can also be applied to real team building. The virtual team only needs telephone and network to coordinate and communicate with each other and even complete the discussion of the project and the sharing of documents; then, a planned job can be performed according to the division of labor [3]. By combining enterprise management with high technology, the organization and management of enterprises can be more scientific and flexible. The virtual team shows a globalization trend and enormous potential. Therefore, increasing organizations find that management based on the virtual team will achieve far-reaching results.

Under the pattern of the knowledge economy, the market and service targets have gradually become the development needs of many enterprises. At this time, the traditional organizational structure can no longer adapt to the complex uncertain factors of the outside world [4]. If the information-sharing mechanism can be used to allow internal members of the enterprise to share their experience and knowledge with others, it will be more conducive to the improvement of corporate organizational capital, thereby creating the core value of the enterprise. To meet the diversified market demands, high-technology-based virtual teams are implementing resource integration across organizations and regions. Summarizing the successful enterprises worldwide in the current environment, virtual teams are an inevitable choice for companies and enterprises to reach a certain stage and respond to market demand [5]. The innovative virtual team model constitutes a flat organizational structure, which is of great significance for enterprises to achieve specific major project goals and overall innovation [6, 7]. Most existing research on virtual teams has no precise project function positioning. In other words, virtual teams are still in the theoretical stage and lack targeted research on practical applications, such as virtual marketing teams and virtual research and development (R&D) teams. Moreover, the research on virtual team management in different task situations and life cycle stages is less involved. Therefore, it is essential to study and design the construction process of the virtual team and its supporting management process and mechanism.

For the knowledge transfer of virtual teams, compared with traditional knowledge transfer, it has its unique management model. The construction of the virtual team is based on dynamic alliances; thus, the composition of team members should be more refined and there are higher requirements for the speed of responding to the market. In this study, based on the neural network (NN), the research and development (R&D) and innovation capabilities of enterprises are studied, and an evaluation index system is constructed. In the stage of empirical analysis, this study selects Enterprise A for research and analyzes the status quo and existing problems of this company. This study puts forward a set of schemes conducive to virtual team cooperation and comprehensively uses management theory to evaluate its application effect. Finally, the team cooperation optimization model is combined with the technological innovation ability supported by the neural network model, which is expected to provide a basis for enterprise knowledge transfer and technological innovation. Compared with existing studies, the new enterprise R&D capability evaluation index system reported here can reflect the R&D process and the factors involved in the R&D process. Besides, the logic of the index system is clearer, and the content is more comprehensive than previous methods.

## 2. Theoretical basis of virtual team collaboration

### 2.1 Virtual team and organization management

Virtual enterprises and organizations are also based on this principle. Through the integration of external resources of the enterprise, the innovation ability and competitiveness of the enterprise are improved, and the enterprise goals are effectively achieved [8]. Virtual enterprises or organizations have surpassed the limits of physical enterprises on the geographical and organizational boundaries, and have continuously expanded their capabilities. As a flexible organization form across regions and departments, virtual teams are represented by flat dynamic networks to realize organization virtualization. It is the preferred solution for quickly sharing knowledge and experience to achieve organizational goals [9]. The emergence of virtual teams has broken the boundaries between enterprises and departments, strengthened the internal cohesion of the team, pursued a mutually beneficial cooperative relationship through mutual learning, and integrated management and technical advantages. Therefore, the application of virtual teams is of great significance to the enterprise, which not only reduces the office costs of team members but also improves the efficiency of information communication between members through advanced communication technologies. Because they are not restricted by regions, virtual teams also have advantages for employees in terms of organizational recruitment. They can eliminate the interference of external factors such as the area of residence, which is more beneficial for enterprises to attract outstanding talents [10]. The most apparent difference between virtual teams and traditional teams is that virtual teams emphasize communication and dynamic cooperation across time, space, and organizational boundaries with the help of network communication technology to complete specific tasks. The management and executive personnel in the virtual team can communicate directly and interact closely.

The organization model of the virtual team is composed of two parts, i.e., the core layer and the peripheral layer. The core layer is composed of several enterprises and is a relatively stable structure. Through an Alliance Steering Committee (ASC), this virtual team performs internal management and coordination, including administrative, technical, financial, and legal affairs. ASC can be regarded as an information communication module. When enterprises subsequently integrate other task modules, ASC still needs to act as a medium for information communication. It connects the peripheral enterprises with the core layer and timely sets up or adjusts work teams according to the status of the organization to complete work tasks [11–13]. The model of virtual team organization structure is shown in Fig 1. In the daily operation of the virtual team, ASC is linked with the coordination module to coordinate the problems in the operation. The coordination module coordinates the internal modules through information sharing. The network structure formed between the ASC and the coordination modules and internal modules of the peripheral enterprises is the basis for ensuring the existence and development of the virtual team and maintaining the normal operation of the team.

Based on the overall structure model of the virtual team, the organization mode of the virtual team can be divided into three types: parallel, star, and federated according to the number of core layer members. In the parallel mode, there is no distinction between the core layer and the peripheral layer, and the members have no ranks. This situation is ideal but basically does not exist in practical applications. The core layer of the star model has only one enterprise; thus, it needs to bear the responsibility of ASC, and the rest of the cooperative enterprises belong to the peripheral layer [14]. The federated model is the most applied virtual team model. The core layer is composed of multiple backbone companies. The partner is selected as the peripheral layer according to the project or market requirements. The specific structure is shown in Fig 2. The integration task module can be jointly undertaken by the cooperative enterprises in the organization or be solely borne by a certain enterprise [15].

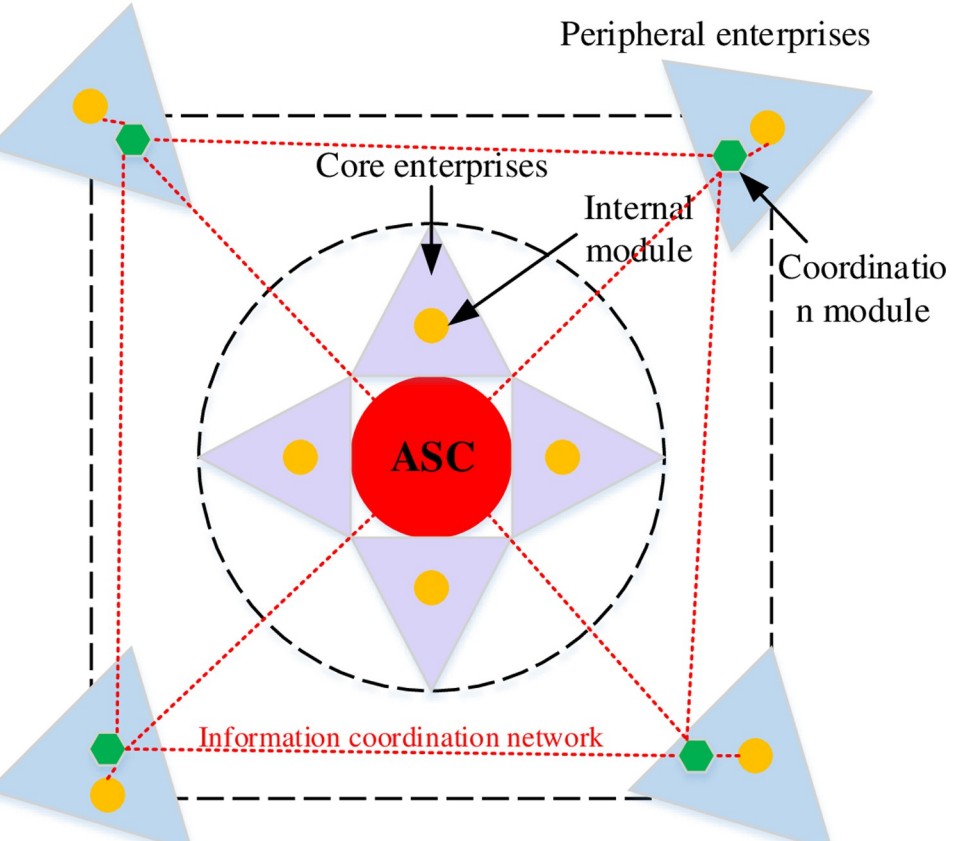

**Fig 1. Model diagram of virtual team organization structure.**

At present, the traditional enterprise team management model pays more attention to mobilizing the enthusiasm of employees and emphasizes a new management model of participatory management. Based on this participatory management method, employees can achieve self-worth in their work, thereby enhancing team cohesion, and finally, being reflected in actual production. A team is a special group that emphasizes collective performance. A well-functioning team must have clear overall goals, an efficient leadership team, complementary members, and smooth communication among members [16]. Large-scale enterprises must promote teamwork to achieve organizational goals. The participation of each employee is the key to overall quality optimization. In the management process of the virtual team, corresponding management should also be carried out according to each stage of the life cycle. The specific management content of the four stages of the gestation period, formation period, operation period, and disintegration period are shown in Table 1.

## 2.2 Virtual team collaboration from the perspective of knowledge transfer

The ability of organizational learning determines the core competitiveness of an enterprise to a certain extent, especially in the current rapidly changing environment of internal and external competition. Only through continuous organizational learning and timely adjustments based on environmental changes can enterprises promote their performance. Organizational learning can improve the ability of enterprises to adapt to uncertain and unknown environments, and knowledge transfer can bring competitive advantages to organizations [17]. Some scholars

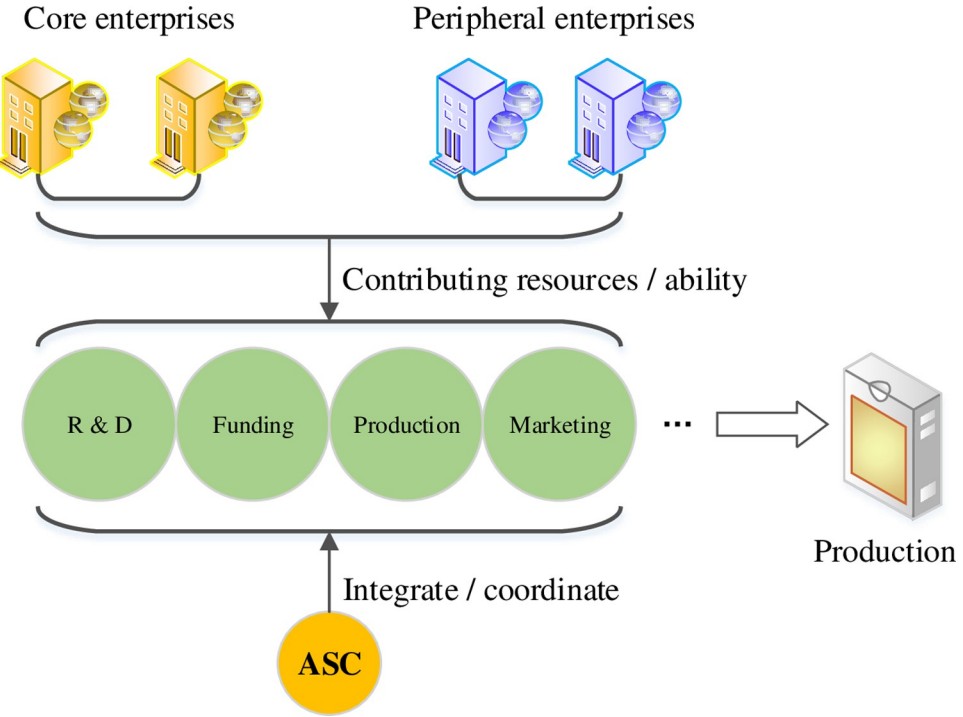

**Fig 2. Virtual team structure in the federated model.**

believe that in highly centralized enterprises, the channels for knowledge transfer are usually single and follow the top-down principle. In a flat organic organization such as a virtual team, the direction of information exchange exists in various forms, and the diversification of information communication channels provides convenience for knowledge dissemination, which is more conducive to the integration and absorption of knowledge by enterprises [18, 19]. The

**Table 1. Specific management content of each period of the virtual team.**

| Period | Management content |
|---|---|
| Incubation | Identification and evaluation of market opportunities |
| | Resource and capacity analysis |
| | Virtual team goal setting |
| | Operation mode selection |
| Formation | Choice of virtual team partners |
| | The choice and determination of cooperation mode of member enterprises |
| | Organization design |
| | Agility measurement |
| | The signing of the cooperation agreement |
| Operation | Partnership management |
| | Task allocation and coordination |
| | Operation supervision and feedback |
| Disintegration | Project suspension identification |
| | Benefit distribution |
| | Transactions during dissolution |
| | Transactions during dissolution |

knowledge within the industry-university-research collaborative innovation team adds value during the gain and drain, and the complementary resources and collaborative development are realized through knowledge transfer. The smooth flow of knowledge among the members determines the size of the industry-university-research collaborative innovation ability. Therefore, knowledge transfer is the essence of industry university research collaborative innovation activities and a necessary procedure of industry-university-research collaborative innovation activities.

For the knowledge transfer of virtual teams, some scholars divide it into five stages, i.e., knowledge search, clarification, flow, understanding, and innovation. The first problem that the knowledge transfer of virtual team solves is to clarify where the knowledge required to complete the target project is and use this as a starting point for the members of the organization to conduct knowledge collection through various channels. After screening, the most relevant knowledge content of the target task is obtained for knowledge circulation, which is also a substantive link to knowledge transfer activities [20]. The characteristics of a virtual team determine that its knowledge transfer process is different from that of a general team. Generally, innovative information technology is required to meet the needs of members' knowledge transfer and information exchange and to coordinate the task allocation among decentralized members. Knowledge transfer within an enterprise or organization needs to follow a certain mechanism framework. An effective and reasonable knowledge transfer system can tap the internal motivation of members and promote a stable and virtuous circle of knowledge transfer within the enterprise. Based on advanced communication technology, task assignment is combined with the characteristics of target projects and members to ensure effective technical collaboration. However, at the same time, due to the differences in technical capabilities between members, it may bring some obstacles to the knowledge transfer process and make the management of the virtual team more complicated [21]. In general, due to the cross-time and cross-regional nature of virtual teams, members need to communicate by phone or network. This form of non-face-to-face communication is difficult to establish trust among members. A large number of studies have found that the construction of virtual teams, the determination of goals, and the arrangement of division of labor need to be based on the mutual trust of members. Therefore, enterprises should establish social interconnection relationships among members as much as possible, use multimedia technology to enrich communication methods, and reduce the obstacles to trust caused by the inability to communicate face to face. Where conditions permit, face-to-face offline work communication or group building activities can be organized to eliminate the gap between members and enhance team trust.

## 3. An empirical study on enterprise capability evaluation based on NN

### 3.1 Evaluation of enterprise technology innovation capability based on the NN model

The NN evaluation method belongs to a key method in the field of artificial intelligence evaluation. By simulating the biological NN processing method, an evaluation model suitable for dealing with real problems is constructed. The NN can store the experience summarized from the samples in the network system in the form of weights. These experiences are then used in the evaluation of performance, ability, and risks.

NN in living organisms is composed of about $10^{12}$ neurons, and each neuron is connected to about 100 neurons to form a giant complex NN. The dendrites of a neuron are responsible for receiving signals, which are processed by the cell body and then transmitted to the synapses

through the axons; then, the signals are transmitted to the next neuron [22]. The artificial neural network (ANN) based on the working principle of the biological NN is a simplified biological NN. The strong self-learning ability, parallel computing ability, and storage capacity of the NN determine that the NN evaluation method also has good robustness and fault tolerance [23]. After several times of learning in the evaluation process, NN finally obtains appropriate weights, which reduces the interference of human factors on the evaluation results and improves the accuracy of the evaluation.

The MATLAB software is used to realize the construction of a visual NN evaluation model. In the input layer, the input variable is set to 12, indicating that the evaluation index system is composed of 12 indexes, and each neuron represents an index. Normalization processing is required before input. The back propagation neural network (BPNN) structure can reflect all continuous functions. The number of neurons in the hidden layer is set to 11. Since the evaluation conclusion is single, the number of neurons in the output layer is 1. There is only one neuron in the output layer. There is no particularly clear method to determine the number of neurons in the hidden layer. If there are too few units in the hidden layer, there will be too little information that the network can obtain to solve problems. Otherwise, the training time will be increased and the learning time will be extensively extended, resulting in the "overfitting". The normalization processing needs to be conducted on the input value of the input layer first before entering the network. The result is obtained by inputting the sorted data into the trained model. In the process of evaluating the technology innovation capability of enterprises, the assimilation processing is first performed on qualitative indicators. Then, the qualitative indicators are normalized along with the quantitative indicators, which are ultimately brought into the evaluation model. The quantitative data is obtained by calculating, which is transformed into a qualitative conclusion by the evaluation set.

## 3.2 Empirical research objects

Enterprise A was established in 2008 with a registered capital of 12 million CNY. It is a company that provides Internet strategy consulting and product promotion and operation services. The current staff of Enterprise A is more than 400 people, who are mainly technical staff, with a total of 305 people and accounting for about 75% of the total number. The annual business volume of Enterprise A is growing steadily at a rate of 250%. Especially, the customization and development services of smart device applications (APPs) have accumulated a large number of customers, which has brought opportunities for further expansion of business scope and enterprise scale. Due to the rapid development of the enterprise, its management blindly pursues returns on benefits but does not pay enough attention to internal management. Therefore, the unstable internal environment caused by the flow of staff has affected market competitiveness and threatened the market position of Enterprise A. The formal survey mainly adopts the combination of convenient sampling and snowball sampling to distribute and recover the questionnaire. On the one hand, the paper questionnaire is distributed by convenient sampling, similar to the pre-survey. On the other hand, snowball sampling is used to distribute electronic questionnaires. The data in this paper are from the public data and research data of enterprises, and the data content is reliable and trustworthy.

## 4. Results and discussions

### 4.1 Analysis of the current situation of Enterprise A

In the development process of Enterprise A, the organizational structure has also changed continuously. In the latest adjusted organizational structure in 2017, the core management team includes the president and vice presidents of finance, operations, administration and business,

who are responsible for making decisions on the strategic affairs of the enterprise. The cooperation between its finance department and the commerce department is more frequent, and the financial department needs to participate in the approval of each stage of project completion. The implementation of the project requires the cooperation of various departments. The commerce department is responsible for receiving orders and transferring the project to the project department after signing the contract. The project department conducts a more detailed demand analysis with the customer based on the initial content of the contract. After the customer confirms the plan, the technical department conducts product development. During the development process, the technical department should coordinate and confirm various issues of product development with the project manager and the customer, and finally, deliver the project. The project department submits the project results to the customer for inspection and confirms that the results can be applied. There are more than 300 employees in Enterprise A, of which more than 80% are technical personnel, with a steady business growth rate of 250% per year. At present, the company has set up branches or operating service centers in Shanghai, Beijing, Guangzhou, Shenzhen, and marketing and technical service stations in 13 cities in China.

The problems in project management include: (1) The business team often ignores the potential risks of the project from the perspective of fast signing, or directly pushes the relevant responsibilities to the subsequent responsible team. In addition, the business team hopes to display a low-cost and high-margin quotation data to attract customers; thus, the data may be fictional, which will lead to a smoother project at the time of signing; however, the hidden risks will be exposed to the development and implementation, causing obstacles to the completion of the project by the implementation team, hinders the progress, and increases the costs. (2) The timeliness of the Internet industry determines that the project has higher requirements for the completion progress. Due to the rigid regulations of the current approval process of Enterprise A, there is a situation of tugging between the various departments for the benefit; eventually, the project is postponed. (3) Enterprise A does not set up a product-oriented quality management team. For the project results completed by the technical department, the internal project manager completes the test, and there is no standard system and quality assurance for the test link. The resulting project quality problems will become a shortcoming in the rapid development of the enterprise. (4) In the implementation process of the project, the departments are usually responsible for their affairs. The entire project lacks a responsible team for overall planning; thus, it is very difficult for each department to coordinate work. Due to the inability to achieve consistent thinking, shirking responsibility and mutual accusations often occur, which not only is detrimental to the formation of team cohesion but also harms project completion.

The above-mentioned problems of Enterprise A have exposed its problems in project management in terms of schedule, cost, quality, and cross-departmental collaboration. For Enterprise A, whose main business is project delivery, it should work in a flat and coordinated organizational form to give the project team a higher degree of freedom. Therefore, Enterprise A needs to make corresponding adjustments to its business management processes and work in a team collaboration model that facilitates knowledge accumulation and transfer, allowing knowledge to flow within the organization, forming productivity, saving costs, and improving the performance.

## 4.2 Optimization scheme for the virtual team collaboration model

From the working process of a knowledge-based virtual team, the cooperation mode of virtual team in the early stage of team operation is mainly synchronous remote cooperation, and the

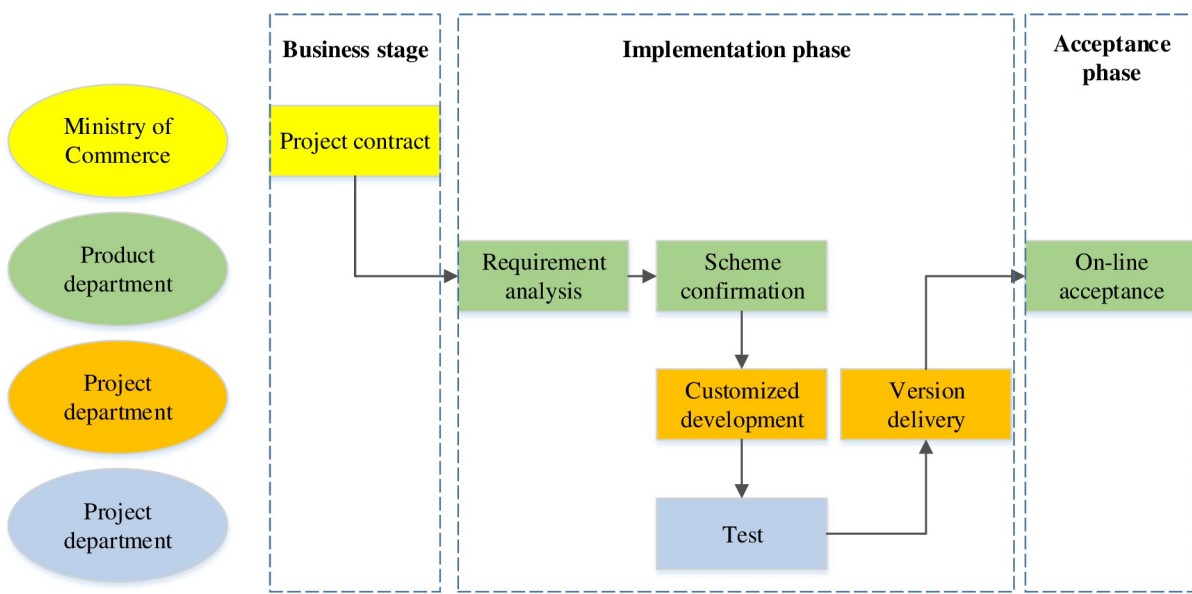

**Fig 3. Optimized team business process.**

corresponding cooperation model is the conference model. During team operation, the cooperation mode of virtual teams is generally asynchronous and remote cooperation, and the corresponding cooperation model is an object-oriented multi-level cooperation model. Before project implementation, the project development team and project manager need to be divided into a unified business department to eliminate communication barriers. In this way, when the project enters the development phase, the manager can directly control the project. In this way, when the project enters the stage after the development phase, the manager's control over the project can be promoted. The new structure should not only focus on the project itself but also be enlarged to the entire enterprise to ensure smooth coordination between projects. When faced with the needs of priority projects, resources can be coordinated across departments to quickly set up a virtual team to complete the task. While adjusting the organizational structure of Enterprise A, the business process was also improved based on the characteristics of the virtual team. The optimized business process is shown in Fig 3. After entering the custom development process, the project is assigned to the project department, and the project manager is responsible for the development progress and cost control of the project. Because the project manager and the development team belong to the project division in the optimized architecture, the collaboration and communication within the team will be significantly improved.

## 4.3 Evaluation of R&D and innovation technology capabilities of virtual teams

For Enterprise A, 75% of its products are independently developed and produced. Therefore, the core technology has a degree of ownership of 0.75. In terms of utilizing the external environment, it has joined 2 technology alliances. This study builds a NN model. The step size in the process of training the network is set to 1000, and the training network reaches the required accuracy after multiple operations. Comparing the simulated data with the expected data, the expected value is reached, indicating that NN can be used for subsequent evaluation. The data of Enterprise A are collected and normalized. The specific results are shown in Table 2 and Fig 4. The processed data are used as the input variable of the NN model.

**Table 2. Values and normalization of each indicator.**

| Number | Index | Enterprise A indicators | Normalization |
|---|---|---|---|
| 1 | Number of R&D personnel | 16 | 0.004 |
| 2 | Although the number of R&D personnel is strong | 0.041 | 0 |
| 3 | Technical background business management director | 0.30 | 0.974 |
| 4 | Technical background, business management strength | 2 | 0.155 |
| 5 | Number of R&D investment | 0.25 | 0.398 |
| 6 | R&D capital per capita of R&D personnel | 9808766.3 | 0.022 |
| 7 | Training and utilization of R&D personnel per capita | 233987.4 | 0.434 |
| 8 | R&D investment intensity | 0 | 0 |
| 9 | Advanced level of R&D equipment | 0.012 | 0.062 |
| 10 | Establishment of R&D institutions | 0.5 | 0.2 |
| 11 | Number of patents | 2 | 0.003 |
| 12 | Number of patents per R&D personnel | 295 | 0.04 |

## 4.4 Selected results of cooperating objects of the virtual team

Enterprise A, as a component of the core layer of the virtual team, needs to continuously absorb cooperative enterprises as alliances in its development. Therefore, it is necessary to analyze the evaluation results of the cooperative enterprises. In this study, risk factor $\varepsilon = 0$ is selected to convert the original interval values of technology candidate enterprises into point values. The results of the original data are shown in Table 3 and Fig 5. According to the membership degree of the candidate enterprise partners, it is known that Enterprise 3 > Enterprise 5 > Enterprise 1 > Enterprise 6 > Enterprise 2 > Enterprise 4. In addition, by selecting different values, the selection results will also change, providing a range of operational preferences for alliance leaders in the virtual team.

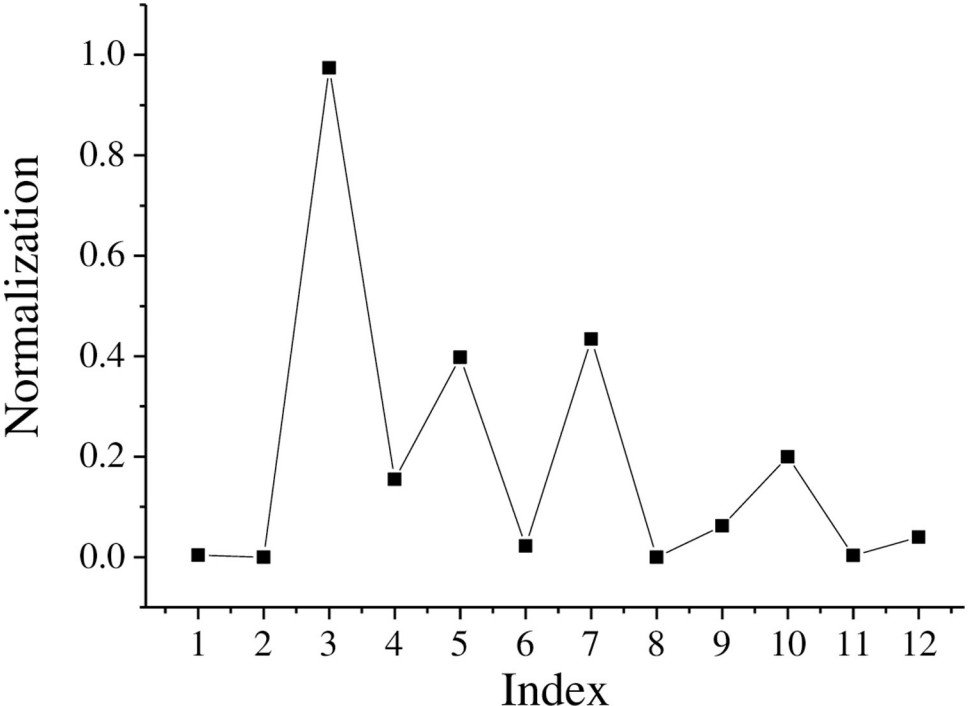

**Fig 4. Normalized results of 12 indicators.**

**Table 3. Post-specification point values for technology candidate enterprises.**

| Enterprise | Cost | Response time to technical requirements | Quality system | Innovation ability |
|---|---|---|---|---|
| 1 | 0.096 | 0.104 | 0.107 | 0.098 |
| 2 | 0.108 | 0.092 | 0.125 | 0.114 |
| 3 | 0.105 | 0.088 | 0.107 | 0.084 |
| 4 | 0.092 | 0.121 | 0.107 | 0.118 |
| 5 | 0.100 | 0.106 | 0.125 | 0.091 |
| 6 | 0.097 | 0.099 | 0.137 | 0.088 |

## 4.5 Discussion of theoretical value and practical significance

The knowledge transfer of organizations is carried out under a certain mechanism framework. Whether effective, open, and transparent knowledge sharing and transfer can be carried out between individuals, and between teams and teams depends on the construction of knowledge transfer mechanism. An effective knowledge transfer mechanism can explore the intrinsic motivation of members, cultivate the atmosphere of organizational sharing, and form a lasting virtuous circle of knowledge transfer within the enterprise. In the knowledge transfer value chain of virtual team, data comes from each node in the network value chain, and the continuous circulation of the value chain integrates the decentralized organization into a compact system. Consequently, the team's knowledge transfer efficiency will increase around this network value chain. The networking of knowledge can optimize the team's cooperation and improve the efficiency of team cooperation, thus providing a guarantee for the team to better complete the task and create new products and services.

Talent is the paramount resource in enterprises, so it is necessary to retain talents. Improving the enthusiasm of talents requires a sound incentive mechanism. The performance of

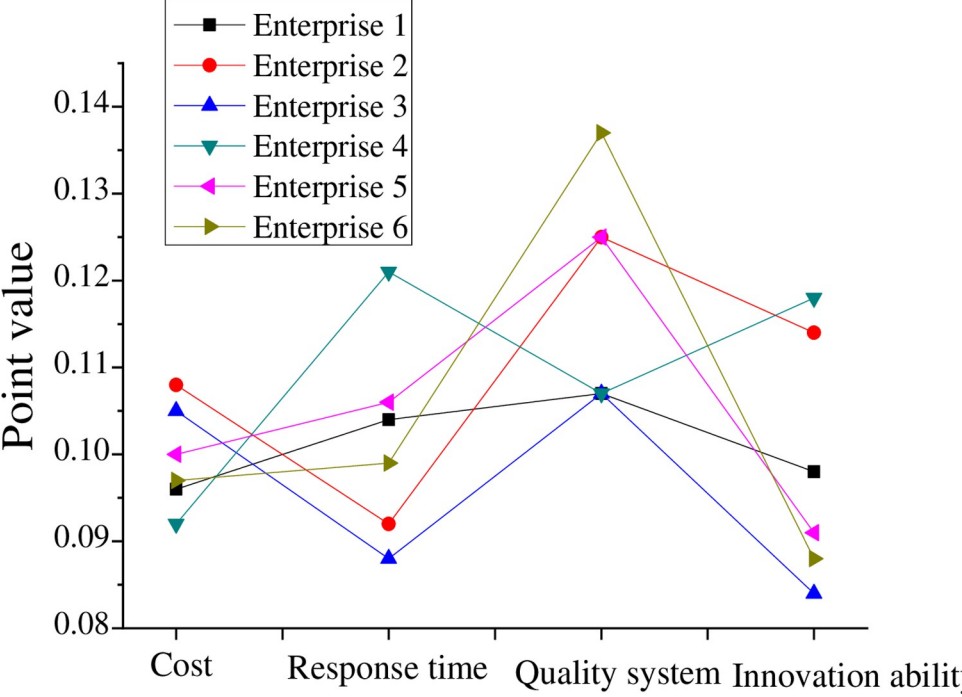

**Fig 5. Comparison of point values of technical candidate enterprises after the specification.**

scientific research personnel should be directly linked to scientific research projects and scientific research achievements, to enable researchers to make more active efforts to complete research and development tasks. Moreover, there should be various forms of reward, both material reward and spiritual reward. The reward mechanism is actually the affirmation and recognition of the contribution of enterprises to R&D personnel, which can empower the rewarded personnel with more passion and motivation.

## 5. Conclusions

A virtual team is a hot topic in the current discussion of management. In the process of operation, the virtual team is affected by both subjective and objective factors. Therefore, it is a complicated process. This study discusses the optimization of organizational management from the perspective of virtual team knowledge transfer and comprehensive use of management theory. Based on the NN, the research and development (R&D) and innovation capabilities of enterprises are studied, and an evaluation index system is constructed. This study selects Enterprise A for research and analyzes the status quo and existing problems of this company. Then, this study proposes a set of schemes conducive to virtual team collaboration and comprehensively uses management theory to evaluate the application effect. During the project implementation process of an enterprise, when faced with the needs of priority projects, resources can be coordinated across departments to quickly set up a virtual team to complete the task. The team collaboration optimization model is combined with the technological innovation capability under the NN model, and it is expected to provide a basis for the knowledge transfer and technology innovation of the enterprises. This paper introduces the concept of virtual teams into R&D teams. Under the flawed theory of virtual R&D team, make full use of the respective theories of R&D team and virtual team, this paper integrates the advantages of traditional R&D team and virtual team into the virtual R&D team and highlights the characteristics and application significance of this flat organization. There are still limitations to this study. As a brand new model, the virtual team still lacks comprehensive understanding and effective control methods in an environment of continuously accelerated market rhythm and economic globalization. Therefore, the subsequent study should incorporate the theory of effective risk control into the discussion scope of virtual teams and improve the application efficiency of virtual teams more comprehensively.

## Supporting information

**S1 Data.**
(XLSX)

## Author Contributions

**Conceptualization:** Anqi Zhang.

**Data curation:** Anqi Zhang.

**Formal analysis:** Anqi Zhang.

**Funding acquisition:** Anqi Zhang.

**Investigation:** Anqi Zhang.

**Methodology:** Anqi Zhang.

**Project administration:** Anqi Zhang.

**Resources:** Anqi Zhang.

**Software:** Anqi Zhang.

**Supervision:** Anqi Zhang.

**Validation:** Anqi Zhang.

**Visualization:** Anqi Zhang.

**Writing – original draft:** Anqi Zhang.

**Writing – review & editing:** Anqi Zhang.

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
