## [Decision Letter · Decision Letter 0]

12 Jul 2021

PONE-D-20-35199

The Application of Virtual Teams in the Improvement of Enterprise Management Capability from the Perspective of Knowledge Transfer

PLOS ONE

Dear Dr. Zhang,

Thank you for submitting your manuscript to PLOS ONE. After careful consideration, we feel that it has merit but does not fully meet PLOS ONE’s publication criteria as it currently stands. Therefore, we invite you to submit a revised version of the manuscript that addresses the points raised during the review process.

We look forward to receiving your revised manuscript.

Kind regards,

Alessandro Margherita

Academic Editor

PLOS ONE

Journal Requirements:

3. Please ensure that you include a title page within your main document. You should list all authors and all affiliations as per our author instructions and clearly indicate the corresponding author.

Reviewers' comments:

Reviewer's Responses to Questions

**Comments to the Author**

1. Is the manuscript technically sound, and do the data support the conclusions?

Reviewer #1: No

Reviewer #2: Partly

2. Has the statistical analysis been performed appropriately and rigorously? 

Reviewer #1: I Don't Know

Reviewer #2: Yes

3. Have the authors made all data underlying the findings in their manuscript fully available?

Reviewer #1: No

Reviewer #2: No

4. Is the manuscript presented in an intelligible fashion and written in standard English?

Reviewer #1: No

Reviewer #2: Yes

5. Review Comments to the Author

Reviewer #1: This article covers the importance of virtual teams and the process of knowledge transfer. The article presents original ideas, for example, the representation of virtual teams as flat dynamic networks.

Overall, the article could be better organized. The sections of the text need to be more developed and some of the sentences are not completely clear. For example, “The term “virtual” … refers to technology that does not exist but has physical functions.”

Some ideas need clarification -- “Before the project is implemented, the project development team and the project manager need to be divided into a large business department to eliminate communication barriers.” Is this about a larger team or a division into smaller teams? Is the goal to divide or expand? What exactly is the paper suggesting as a solution?

Regarding the Evaluation of R&D and innovation technology capabilities of virtual teams, the data used for the neural network is not available. The process of using a neural network is not described in detail. Data samples used to train the neutral network are missing. Six enterprises are listed in the “Evaluation of R&D and innovation technology capabilities of virtual teams”. There is no explanation of the choice of indexes used for the six enterprises. The methodology is difficult to analyze. Some common-sense indexes such as experience in the field, references from prior work or trust which could greatly influence a ranking of the enterprises are not mentioned in the evaluation. In conclusion, I think that a comparison between the case model using a neural network and the case model without the use of a neural network could help the reader understand the advantages of using neural networks for the evaluation.

Reviewer #2: The authors should:

1) define more clearly the research gap (in the extant literature) already in the introduction

2) present in a more comprehensive way (e.g. using a chart/process) the research roadmap undertaken

3) distinguish better what, in section 2, represents theory background and which one are arguments/speculations deriving from the same

4) provide more background on the analysed case and justify selection of the same

5) build a section with clear theoretical advancements and practitioner implications of the study

6. PLOS authors have the option to publish the peer review history of their article (what does this mean?). If published, this will include your full peer review and any attached files.

Reviewer #1: No

Reviewer #2: No

---

## [Author Response · Author response to Decision Letter 0]

9 Aug 2021

PONE-D-20-35199

The Application of Virtual Teams in the Improvement of Enterprise Management Capability from the Perspective of Knowledge Transfer

PLOS ONE

Dear Dr. Zhang,

Thank you for submitting your manuscript to PLOS ONE. After careful consideration, we feel that it has merit but does not fully meet PLOS ONE’s publication criteria as it currently stands. Therefore, we invite you to submit a revised version of the manuscript that addresses the points raised during the review process.

If applicable, we recommend that you deposit your laboratory protocols in protocols.io to enhance the reproducibility of your results. Protocols.io assigns your protocol its own identifier (DOI) so that it can be cited independently in the future. For instructions see:http://journals.plos.org/plosone/s/submission-guidelines#loc-laboratory-protocols. Additionally, PLOS ONE offers an option for publishing peer-reviewed Lab Protocol articles, which describe protocols hosted on protocols.io. Read more information on sharing protocols athttps://plos.org/protocols?utm_medium=editorial-email&utm_source=authorletters&utm_campaign=protocols.

We look forward to receiving your revised manuscript.

Kind regards,

Alessandro Margherita

Academic Editor

PLOS ONE

Journal Requirements:

Reply: We have referred to the links you provided to ensure that our manuscript meets the style requirements of PLoS One.

2. PLOS requires an ORCID iD for the corresponding author in Editorial Manager on papers submitted after December 6th, 2016. Please ensure that you have an ORCID iD and that it is validated in Editorial Manager. To do this, go to ‘Update my Information’ (in the upper left-hand corner of the main menu), and click on the Fetch/Validate link next to the ORCID field. This will take you to the ORCID site and allow you to create a new iD or authenticate a pre-existing iD in Editorial Manager. Please see the following video for instructions on linking an ORCID iD to your Editorial Manager account:https://www.youtube.com/watch?v=_xcclfuvtxQ

 Reply: We ensure that we have the ORCID and verify it with the Editorial Manager.

3. Please ensure that you include a title page within your main document. You should list all authors and all affiliations as per our author instructions and clearly indicate the corresponding author.

Reply: We have confirmed that the title page is included in the main document and the author information is clear.

 Reply: We have confirmed that the title of the submission on the Internet is consistent with that in the manuscript.

Reviewers' comments:

Reviewer's Responses to Questions

Comments to the Author

1. Is the manuscript technically sound, and do the data support the conclusions?

Reviewer #1: No

Reviewer #2: Partly

Reply: Thanks for your review and comments to ensure that the results of this study support the corresponding conclusions.

2. Has the statistical analysis been performed appropriately and rigorously?

Reviewer #1: I Don't Know

Reviewer #2: Yes

Reply: Thanks for your recognition of this study. The statistical analysis method in this paper is suitable for the evaluation of technological innovation ability of enterprises.

3. Have the authors made all data underlying the findings in their manuscript fully available?

Reviewer #1: No

Reviewer #2: No

Reply: Thanks for the suggestion. The data in this study are all real survey data, so the data information is true and reliable.

4. Is the manuscript presented in an intelligible fashion and written in standard English?

Reviewer #1: No

Reviewer #2: Yes

Reply: Thank you very much for your review and comments.

5. Review Comments to the Author

Reviewer #1: This article covers the importance of virtual teams and the process of knowledge transfer. The article presents original ideas, for example, the representation of virtual teams as flat dynamic networks.

Overall, the article could be better organized. The sections of the text need to be more developed and some of the sentences are not completely clear. For example, “The term “virtual” … refers to technology that does not exist but has physical functions.”

Reply: Thanks for the suggestion. According to your suggestion, we have optimized the whole article and improved the expression of sentences.

Some ideas need clarification -- “Before the project is implemented, the project development team and the project manager need to be divided into a large business department to eliminate communication barriers.” Is this about a larger team or a division into smaller teams? Is the goal to divide or expand? What exactly is the paper suggesting as a solution?

Reply: In response to your questions, our explanation is: to extract project managers from different departments, build a more comprehensive department, so that the information barriers between departments can be eliminated. The purpose of this is to achieve information exchange and cross sectoral cooperation.

Regarding the Evaluation of R&D and innovation technology capabilities of virtual teams, the data used for the neural network is not available. The process of using a neural network is not described in detail. Data samples used to train the neutral network are missing. Six enterprises are listed in the “Evaluation of R&D and innovation technology capabilities of virtual teams”. There is no explanation of the choice of indexes used for the six enterprises. The methodology is difficult to analyze. Some common-sense indexes such as experience in the field, references from prior work or trust which could greatly influence a ranking of the enterprises are not mentioned in the evaluation. In conclusion, I think that a comparison between the case model using a neural network and the case model without the use of a neural network could help the reader understand the advantages of using neural networks for the evaluation.

Reply: Thanks for the careful reading. We have refined the application of neural network according to your suggestion, and added a more detailed explanation, including the corresponding data input in neural network.

Reviewer #2: The authors should:

1) define more clearly the research gap (in the extant literature) already in the introduction

Reply: Thanks for the careful reading. We have improved the introduction to emphasize the differences between this study and the existing literature.

2) present in a more comprehensive way (e.g. using a chart/process) the research roadmap undertaken

Reply: The research process of this paper is carried out according to the order of each section, which has been clearly reflected in the paper.

3) distinguish better what, in section 2, represents theory background and which one are arguments/speculations deriving from the same

Reply: Those are great suggestions. The second section of the original content has been divided, the theoretical basis and research methods are described separately.

4) provide more background on the analysed case and justify selection of the same

Reply: Thanks for the careful reading. We have added more detailed information to this case study.

5) build a section with clear theoretical advancements and practitioner implications of the study

Reply: Those are great suggestions. We have added a separate section to analyze the theoretical and practical significance of this study.

6. PLOS authors have the option to publish the peer review history of their article (what does this mean?). If published, this will include your full peer review and any attached files.

Do you want your identity to be public for this peer review? For information about this choice, including consent withdrawal, please see our Privacy Policy.

Reviewer #1: No

Reviewer #2: No

---

## [Editor Report · Decision Letter 1]

27 Sep 2021

PONE-D-20-35199R1

The Application of Virtual Teams in the Improvement of Enterprise Management Capability from the Perspective of Knowledge Transfer

PLOS ONE

Dear Dr. Zhang,

Thank you for submitting your manuscript to PLOS ONE. After careful consideration, we feel that it has merit but does not fully meet PLOS ONE’s publication criteria as it currently stands. Therefore, we invite you to submit a revised version of the manuscript that addresses the points raised during the review process.

It is not really clear at what extent the authors have addressed all the comments provided by the two reviewers. If we look at the amendments made in the revised text (i.e. the red coloured parts), it seems that the revisions are quite limited for a double "major revision". Also the explanations provided by the authors are not enough detailed to let understand where and how they brought changes to the original version. The authors are thus required to go once again through all the reviewers' comments and to provide a more extended and convincing discussion of the revision work undertaken.

We look forward to receiving your revised manuscript.

Kind regards,

Alessandro Margherita

Academic Editor

PLOS ONE
---

## [Author Response · Author response to Decision Letter 1]

19 Jan 2022

I have uploaded the latest revised manuscript, please check it！

---

## [Editor Report · Decision Letter 2]

10 Feb 2022

The Application of Virtual Teams in the Improvement of Enterprise Management Capability from the Perspective of Knowledge Transfer

PONE-D-20-35199R2

Dear Dr. Zhang,

We’re pleased to inform you that your manuscript has been judged scientifically suitable for publication and will be formally accepted for publication once it meets all outstanding technical requirements.

Kind regards,

Alessandro Margherita

Academic Editor

PLOS ONE

---

## [Editor Report · Acceptance letter]

15 Mar 2022

PONE-D-20-35199R2 

The Application of Virtual Teams in the Improvement of Enterprise Management Capability from the Perspective of Knowledge Transfer

Dear Dr. Zhang:

I'm pleased to inform you that your manuscript has been deemed suitable for publication in PLOS ONE. Congratulations! Your manuscript is now with our production department. 

Kind regards, 

on behalf of

Dr. Alessandro Margherita 

Academic Editor

PLOS ONE